# Metabolomics Markers of COVID-19 Are Dependent on Collection Wave

**DOI:** 10.3390/metabo12080713

**Published:** 2022-07-30

**Authors:** Holly-May Lewis, Yufan Liu, Cecile F. Frampas, Katie Longman, Matt Spick, Alexander Stewart, Emma Sinclair, Nora Kasar, Danni Greener, Anthony D. Whetton, Perdita E. Barran, Tao Chen, Deborah Dunn-Walters, Debra J. Skene, Melanie J. Bailey

**Affiliations:** 1Faculty of Engineering and Physical Sciences, University of Surrey, Guildford GU2 7XH, UK; may.lewis@surrey.ac.uk (H.-M.L.); yufan.liu@surrey.ac.uk (Y.L.); c.frampas@surrey.ac.uk (C.F.F.); k.longman@surrey.ac.uk (K.L.); m.spick@surrey.ac.uk (M.S.); t.chen@surrey.ac.uk (T.C.); 2Faculty of Health and Medical Sciences, University of Surrey, Guildford GU2 7XH, UK; alexander.stewart@surrey.ac.uk (A.S.); e.l.sinclair@surrey.ac.uk (E.S.); nk00501@surrey.ac.uk (N.K.); a.whetton@surrey.ac.uk (A.D.W.); d.dunn-walters@surrey.ac.uk (D.D.-W.); d.skene@surrey.ac.uk (D.J.S.); 3Frimley Park Hospital, Frimley Health NHS Trust, Camberley GU16 7UJ, UK; danni.greener@nhs.net; 4Manchester Institute of Biotechnology, University of Manchester, Manchester M1 7DN, UK; perdita.barran@manchester.ac.uk

**Keywords:** COVID-19, targeted metabolomics, LC-MS, machine learning

## Abstract

The effect of COVID-19 infection on the human metabolome has been widely reported, but to date all such studies have focused on a single wave of infection. COVID-19 has generated numerous waves of disease with different clinical presentations, and therefore it is pertinent to explore whether metabolic disturbance changes accordingly, to gain a better understanding of its impact on host metabolism and enable better treatments. This work used a targeted metabolomics platform (Biocrates Life Sciences) to analyze the serum of 164 hospitalized patients, 123 with confirmed positive COVID-19 RT-PCR tests and 41 providing negative tests, across two waves of infection. Seven COVID-19-positive patients also provided longitudinal samples 2–7 months after infection. Changes to metabolites and lipids between positive and negative patients were found to be dependent on collection wave. A machine learning model identified six metabolites that were robust in diagnosing positive patients across both waves of infection: TG (22:1_32:5), TG (18:0_36:3), glutamic acid (Glu), glycolithocholic acid (GLCA), aspartic acid (Asp) and methionine sulfoxide (Met-SO), with an accuracy of 91%. Although some metabolites (TG (18:0_36:3) and Asp) returned to normal after infection, glutamic acid was still dysregulated in the longitudinal samples. This work demonstrates, for the first time, that metabolic dysregulation has partially changed over the course of the pandemic, reflecting changes in variants, clinical presentation and treatment regimes. It also shows that some metabolic changes are robust across waves, and these can differentiate COVID-19-positive individuals from controls in a hospital setting. This research also supports the hypothesis that some metabolic pathways are disrupted several months after COVID-19 infection.

## 1. Introduction

COVID-19 was first reported to the World Health Organization (WHO) on 31 December 2019 and has since caused a major global health crisis [1]. Whilst there were rapid advances in both treatment regimens and vaccination development, as of 27 July 2022, there have been over 567 million confirmed cases and over 6.3 million deaths reported globally [2,3,4].

The clinical spectrum of infection for SARS-CoV-2 is broad; severity ranges from asymptomatic to critical illness and mortality [5]. Certain cohorts and patients with comorbidities have been associated with COVID-19-related death [6]. For example, factors such as age, asthma, diabetes, obesity and heart disease put a patient at a higher risk of developing severe disease [6]. The association between COVID-19 and comorbidities such as obesity, diabetes and older age with outcomes suggests that metabolic disturbances may be significant [7]. There is therefore a continuing need to better understand the impact of COVID-19 on the host metabolism, to understand the different clinical presentations and enable better treatments for those affected [8].

Metabolomics is the process of profiling small molecules in order to establish a metabolic profile that directly reflects clinical disturbance [7,9]. Most biomarker research uses blood as a sampling matrix as it is both rich in metabolites and is homeostatically regulated [10]; and as a result, blood-based ‘omics has been shown to outperform other sampling matrices for COVID-19 diagnostics [11]. Metabolic profiling therefore can identify biomarkers and unlike RT-PCR, can be used as both a diagnostic and prognostic tool which will be critical for future epidemics [12].

Numerous studies have now used metabolomics approaches to elucidate the metabolic disturbances caused by COVID-19 infection [7,13,14,15,16,17,18]. For example, the ratio of kynurenine and tryptophan has been found to increase in COVID-19 positive patients [14,17]. Similarly, elevated arginine/ornithine ratios have been found in COVID-19 patients, scaling with severity, indicative of dysregulation in the urea cycle [17,19]. Glutamine metabolism has been found to be altered in positive patients with glutamine/glutamate ratios being seen to be reduced and increased glutamic acid levels [14,20,21]. It has also been reported that lipids involved in glycerol metabolism, particularly triglycerides, are upregulated in positive patients and have a positive correlation with IL-6, a pro-inflammatory protein [13,16,17,18]. Bile acids have also been shown to be dysregulated in COVID-19-positive patients and concentrations were reported to decrease compared to negative patients [22].

Whilst previous work has shed insight into metabolic changes upon COVID infection; the virus, its symptoms, and its clinical management have all evolved over time due to the different emerging variants [23,24]. Our hypothesis is that biomarkers of COVID-19 infection therefore had a time dependence, and this will be of paramount importance for clinical translation. Globally there have been multiple waves of infection, and in 2020, most countries saw a two-wave pattern of COVID-19 disease [25]. In the UK for example, an initial wave of infections from March to June 2020 was followed by a second wave that began in July 2020 as social distancing restrictions were eased [25,26]. In this work, we investigate metabolomic changes associated with COVID-19 infection in the first and second wave of infection in the UK. This is with the goal of both determining the extent to which changes in the virus, its symptoms and treatment have confounded metabolomic analysis, and also identifying biomarkers that are robust in diagnosing COVID-19 in different waves or under different conditions.

## 2. Methods

### 2.1. Participant Sample Collection and Ethics

Ethical approval for this project (IRAS project ID 155921) was obtained via the NHS Health Research Authority (REC reference: 14/LO/1221). A total of 164 participants were recruited at NHS Frimley NHS Foundation Trust hospitals by researchers from the University of Surrey, Frimley Park Hospital and Wexham Park Hospital. Patients were recruited between May 2020 and August 2021 and were separated out into two “waves”: Wave 1 between May 2020 and July 2020, and Wave 2 between September 2020 and June 2021. Participants were identified by clinical staff to ensure that they had the capacity to consent to the study and were asked to sign an Informed Consent Form based on the International Severe Acute Respiratory and emerging Infection Consortium/World Health Organization (ISARIC/WHO) Clinical Characterization Protocol for Severe Emerging Infections. Those patients that did not have this capacity or who did not sign the form were not sampled. The signatures were witnessed by University of Surrey researchers or but hospital research staff. Participants were recruited if there was clinical suspicion of COVID-19 infection. Inclusion for participants was determined by their reverse transcription polymerase chain reaction (RT-PCR) results. Therefore, only patients with confirmed positive or negative RT-PCR tests were included (no inconclusive). All participants were provided with a Patient Information Sheet explaining the goals of the study.

Of the 164 patients, 7 of those returned to supply longitudinal samples. The timing of the longitudinal sample varied from 2 to 7 months after the first hospital sample was taken, as shown in Appendix A. Serum was also taken from 15 healthy controls who were not infected (and had no reported prior infection) with COVID-19. These controls were not age-matched with the patient population and had an average age of approximately 25 compared to the patient population average of 62 (see Table 1).

### 2.2. Metadata Collection

The patients were given an individual ID number to protect their identity. Metadata were collected for each participant and was documented. Data were collected for age, gender, ethnicity, body mass index (BMI), pre-admission symptoms and duration, time between RT-PCR test and sampling, comorbidities, regular and treatment medication, vaccine status, recent blood results, bilateral chest X-ray results, oxygen requirement, intensive care admission and whether the patient survived admission.

### 2.3. Targeted Metabolomics

The solvents used to prepare the solutions and mobile phases: methanol (MeOH), acetonitrile (ACN), water (H_2_O), isopropanol (IPA) and formic acid (FA) were Optima^TM^ LC-MS grade (Fischer Scientific).

Serum collection and extraction followed the protocols set out by the COVID-19 Coalition [27]. Patient blood was collected in 3 mL serum tubes and was transported to University of Surrey by courier whilst stored on ice. All samples with a time interval greater than four hours between collection and extraction were rejected. On arrival, the samples were centrifuged at 1600× *g* for 10 min at 4 °C. Serum was then decanted into 100 µL aliquots and stored at −80 °C until processing.

Prior to analysis, the inactivation of SARS-CoV-2 was undertaken by adding 200 µL ethanol into 100 µL of serum (2:1 *v*/*v* solvent/sample ratio) and vortexed for 20 s as based on a protocol recommended by the World Health Organization (WHO) [28]. The inactivated serum samples were analyzed using the MxP Quant 500 system (Biocrates Life Sciences AG, Innsbruck, Austria) following the protocol of the manufacturer. The MxP Quant 500 system provides targeted quantification of up to 630 metabolites using tandem mass spectrometry (MS/MS).

Sample order was randomized across each plate and 30 µL of inactivated serum was added to a 96-well filter plate containing isotopically labelled internal standards. Additionally, 7-point calibration standards (10 µL), 3 repeat phosphate-buffered saline (PBS) for blank correction and 3 levels of quality controls (QC) were run on each plate. The filter plate was dried under nitrogen and a phenyl isocyanate (PICT) derivatization reagent was added to each well. After an incubation period and further drying under nitrogen, analytes were extracted using ammonium acetate in methanol (5 mmol/L). The resulting elute was then divided between two plates, one for liquid chromatography mass spectrometry (LC-MS) analysis and one for flow injection analysis (FIA) and further diluted using defined volumes.

Analysis was undertaken using a Xevo TQ-S Triple Quadrupole Mass Spectrometer coupled to an Acquity UPLC system (Waters Corporation, Milford, MA, USA). Each sample measurement required two UHPLC runs (positive and negative mode) and three FIA runs (positive ion mode with varying mass ranges). Chromatographic separation was accomplished using a C18 reversed phase column (supplied by Biocrates) using two mobile phases, solvent A: Water with 0.2% formic acid and solvent B: acetonitrile with 0.2% formic acid using a gradient profile detailed by the manufacturer protocol. The FIA run used one solvent, which was methanol with a modifier provided by the manufacturer.

Biogenic amines and amino acids were quantified for each plate using a seven-point calibration curve, with other analytes semi-quantitated with a single point standard. The levels of metabolites present in each QC were compared to the expected values and the CV% calculated. Data were normalized between the three batches using the results of quality control level 2 (QC2) repeats across the plate (*n*  =  5) and between plates (*n*  =  3) using Biocrates METIDQ software (QC2 correction). Metabolites for which >25% of samples returned concentrations at or below the limit of detection (<LOD) were excluded (totaling *n* = 150). The remaining 474 quantified metabolites comprised of 8 acylcarnitines, 20 amino acids, 26 biogenic amines, 11 bile acids, 53 ceramides, 15 cholesteryl esters, 1 cresol, 9 diglycerides, 4 carboxylic and fatty acids, 85 phosphatidyl cholines, 14 sphingolipids, 222 triglycerides, 2 hormones, 2 indoles, 1 nucleobase and 1 vitamin.

### 2.4. Exploratory Data Analysis

Initially, the data were investigated by multivariate analysis, using both principal component analysis (PCA) and orthogonalized partial least squares discriminant analysis (OPLS-DA). These steps were performed using SIMCA-17 software (Umetrics, Sweden).

For the initial pre-processing step, the data were log transformed, pareto scaled and mean-centered to reduce skew in the data and assume normal distribution. The quality of each model was assessed using the variation between classes (R2Y) and the predictive ability (Q2Y), where the difference between the two values indicates the goodness of fit. When Q2Y > 0.5, the separation can be considered “good”, and when Q2Y > 0.9 there is “excellent” separation [29]. A trustworthy model should see a difference between R2Y and Q2Y as <0.3 [29]. Variable importance in projection (VIP) scores were used to measure the importance of the metabolites with respect to classification in the OPLS-DA model.

### 2.5. Machine Learning Model Construction

Following exploratory data analysis, a machine learning approach was used to investigate the unique biomarkers in each wave, and also to find common biomarkers across waves. Machine learning model construction consists of five major modules, all of which must be present for robust model construction. These modules are: dataset construction, dataset splitting, feature selection, model selection and hyperparameter tuning, and finally model fitting and assessment.

As the first step, 3 datasets were constructed: the first contained positive patients from Wave 1 and negative controls, the second consisted of positive patients from Wave 2 and negative controls, and the third included positive patients from Wave 1 and Wave 2 as well as negative controls. Second, prior to the application of machine learning algorithms, sample set partitioning based on a joint x–y distances (SPXY) algorithm was performed in order to split the dataset into a training set and test set. Compared to other common dataset splitting methods, SPXY, an algorithm based on the Kennard-Stone (KS) algorithm, has been shown to select more representative samples, which is important for reliable model building [30,31]. This step is also important in preventing overfitting of models to the data, by building the model on the train set, and then testing its performance on unseen data, i.e., the test set, for validation [30,31,32]. After train/test split, the data were standard scaled for the pre-processing step. Third, the 30 metabolites with the highest VIP scores obtained from the exploratory data analysis stage were used at the feature selection stage. As metabolites with higher VIP scores are more likely to be relevant to classification these 30 metabolites were used to improve computing power efficiency. Recursive Feature Elimination (RFE, provided by Scikit-learn package) was then applied to these 30 high VIP features to the datasets containing both Wave 1 and Wave 2 patients [33]. This was carried out to further reduce the number of features from the initial 30 metabolites to find the optimal set of metabolites that can be used as potential biomarkers in Wave 1 and Wave 2 datasets.

After feature selection, the fourth step was model selection and hyperparameter tuning and the GridsearchCV algorithm was implemented in this stage. GridsearchCV is a module provided by the Scikit-learn package for searching for the best hyperparameter combination and verifying it by means of cross-validation. In this work, a search space containing 4 learning algorithms: K-neighbors classifier, Logistic Regression, Random Forest Classifier and Decision Tree Classifier was defined. The corresponding hyperparameters are shown in Appendix A. Finally, the selected model and its hyperparameters were used to fit the training set and the performance on the test set was evaluated by the means of accuracy, sensitivity, specificity and F1 score. Confidence Intervals (CI) of accuracy and specificity were calculated based on the assumption that the test accuracy is Gaussian distributed. CI of sensitivity were calculated by normal approximation interval with continuity correction. The model with the highest accuracy was assigned as the diagnostic model in each dataset [34].

## 3. Results

### 3.1. Metadata Summary

The study population consisted of 164 participants, 41 patients with a negative COVID-19 RT-PCR test, but with clinical COVID-19 symptoms, and 123 patients with a positive RT-PCR test. The positive patients were also split into the two collection waves with Wave 1 patients (*n* = 32) recruited between May 2020 and July 2020, and Wave 2 patients (*n* = 91) collected between September 2020 and June 2021. A summary of the metadata comparing positive and negative patients, as well as Wave 1 and Wave 2 positive patients is shown in Table 1. Independent two-sided *t*-tests (95%) were used to determine significant differences.

COVID-19-positive and -negative patients were closely matched in age (mean of 62.4 and 61.6 years, respectively) and the positive patients were age matched between Wave 1 and Wave 2 (mean of 61.7 and 61.8, respectively). There was a higher percentage of male patients in COVID-19-positive patients compared to the negative patients, but the gender profiles for Wave 1 and Wave 2 COVID-19-positive patients were similar. Comorbidities (hypertension, high cholesterol, Type 2 diabetes and heart disease), as well as smoking status were closely represented between positive and negative patients and between waves (*p* > 0.05). COVID-19-positive patients were more likely to be admitted to the Medical Acute Dependency Unit (MADU) and Intensive Care Unit (ICU), to require oxygen and continuous positive airway pressure (CPAP) and to present with bilateral lung X-ray changes, low lymphocytes and eosinophils compared to negative patients. However, the C-reactive protein (CRP) values were similar between positive and negative patients.

Comparing Wave 1 and 2 positive patients, lymphocytes were significantly higher in Wave 2 whereas the CRP and eosinophil levels were significantly higher in Wave 1. This may be due to the increased steroid treatment regime in Wave 2 [35]. Dexamethasone is a steroid hormone which started to be introduced as a COVID-19 treatment for those patients that required oxygen support [3,36]. After successful trials, dexamethasone was made available to hospital patients in June 2020. None of the Wave 1 patients had been prescribed dexamethasone (except for one patient with a negative RT-PCR test for other medical reasons) however, 65 of the 94 patients with a positive RT-PCR test in Wave 2 were prescribed dexamethasone.

### 3.2. Metabolic Profiling: Exploratory Data Analysis

Principal component analysis (PCA) of all the samples (*n* = 164) was used to explore dominant sources of variation in the dataset. Many different sub-classes were investigated by color coding the PCA plot including plate number, gender, age and RT-PCR test result. There was no obvious separation evident in the unsupervised PCA plots (Appendix A). Some separation between positive and negative patients was achieved using an orthogonal partial least squares (OPLS-DA) model, as shown in Figure 1A (R2Y = 0.476 and Q2Y = 0.248). In Figure 1A, three patients assigned as “negative” (396, 398 and 430) can be seen to cluster with the positive patients on the right side of the chart. These three patients produced negative RT-PCR tests at the time of sampling but had previously tested positive and were in ICU during their hospital admission (See Appendix A for reclassification of these patients). We explored excluding patients sampled more than 14 days after the positive RT-PCR test, as well as the patients who had previously tested positive. The separation between positive and negative patients, using OPLS-DA is shown in Figure 1B, improved further (R2Y = 0.712 and Q2Y = 0.418).

The patient cohort were also split into collection waves. The OPLS-DA separation between positive and negative patients is shown in Figure 2A (Wave 1) and Figure 2B (Wave 2). Treating the waves separately showed improved separation than when both waves are considered together (Wave 1: R2Y = 0.955 and Q2Y = 0.408, and Wave 2: R2Y = 0.877 and Q2Y = 0.633). The positive patients from Wave 1 and Wave 2 were also compared, as shown in Figure 2C. A cross-validated residuals (CV-ANOVA) test was undertaken to validate the models and the *p* values were calculated as 6.98 × 10^−4^, 4.80 × 10^−20^ and 1.56 × 10^−11^, respectively showing significant differences. The top VIP scores causing the separation for all 3 models are presented alongside each chart in Figure 2.

The population datasets identified as part of the exploratory analysis as well as the high VIP score features were then carried through to the construction of the machine learning diagnosis model. This was performed to find a set of COVID-19 biomarkers that was robust across collection waves.

### 3.3. Metabolic Profiling: Machine Learning-Based Diagnosis Model

As set out in Methods, a search space comprising learning algorithms: K-neighbors classifier, Logistic Regression, Random Forest Classifier and Decision Tree Classifier was defined. The Random Forest classifier model performed the best across the waves when assessed by diagnostic accuracy, and this classifier model was carried forward for analysis in this study.

Figure 3A shows the most predictive feature sets identified by machine learning for each dataset, and highlights the similarities and differences between the different waves. The Venn Diagram shows that there was no overlap between Wave 1 and Wave 2, but the “Two waves” dataset included some features from each of the individual waves, and also some additional features with good discriminatory power for the overall population. Figure 3B shows the number of potential COVID-19 biomarkers identified by a Random Forest Classifier model. The model identified different features for Wave 1 and Wave 2, as well as 6 features as being diagnostic across waves.

Boxplots for the 6 biomarkers shown to differentiate positive and negative patients across both waves are shown in Figure 4. All six were shown to be significantly different between the total (Wave 1 and Wave 2) positive and negative patients (*p* < 0.05 confirmed using Mann–Whitney U test).

These 6 metabolites were also examined in the longitudinal samples donated by a smaller subset of patients. Figure 5 compares the first hospital sample (Day 0), the average of the samples taken in hospital (Day 0 and Day 2), and the longitudinal sample taken 2–7 months later. These values were also compared to the COVID-19 negatives and the healthy controls. Figure 5 shows that TG (18:0_36:3) and Asp were elevated during hospitalisation, but on follow-up, the levels dropped, and became consistent with the negative patients and healthy controls. In contrast, glutamic acid, found to be elevated in COVID-19 positive patients remained elevated, even on follow-up. The other three metabolites showed no significant difference across the 5 categories (see Appendix A).

## 4. Discussion

This work shows for the first time that the metabolic dysregulation due to COVID-19 is dependent on the wave of infection. Splitting the patients into two collection waves provided considerably better separation between positive and negative patients. For Wave 1 patients, COVID-19 infection was found to be associated with triglycerides and two bile acids: glycolithocholic acid (GLCA) and glycolithocholic acid sulphate (GLCAS); Figure 2A. The triglycerides were upregulated, and the bile acids downregulated. This observation is consistent with previous reports that triglycerides are dysregulated in COVID-19 patients, particularly in patients with severe disease, which is likely because COVID-19 causes an inflammatory response due to the so called cytokine storm which disturbs lipid metabolism pathways [13]. During autophagy, where the body digests cellular components thereby recycling specific biochemicals, free fatty acids are converted into triglycerides [37]. Bile acids have also been previously reported to be down-regulated in COVID-19 patients [22,38]. These data provide support to the hypothesis that, in the first wave of infection, the gut microbiome milieu and content was disrupted in positive patients, as seen in primates infected with COVID-19 [39,40].

However, comparison of Wave 2 positives with the negative controls shows good separation (Figure 2B), arising from a different list of metabolites. Similar to Wave 1, triglycerides were upregulated in positive patients, but the separation results from a different list of triglycerides. Two triglycerides were common to both Wave 1 and Wave 2: TG (20:1_32:3) and TG (22:4_32:2). In addition, aspartate (Asp) and methionine sulfoxide (Met-SO) were upregulated in positive patients in Wave 2. Interestingly, the bile acids, which were found to separate the Wave 1 patients from the negatives, were not included in the top VIP scores for Wave 2.

Additionally, there was found to be good separation between Wave 1 and Wave 2 COVID-19-positive patients, suggesting that the second wave of infection caused different alterations to the metabolome. Cortisol was found to be downregulated in Wave 2 patients compared to Wave 1 patients (Figure 2C). One explanation for this could be the introduction of Dexamethasone as a treatment regime in Wave 2. Dexamethasone is a corticosteroid and causes the suppression of cortisol [41]. However, not all Wave 2 patients were treated with Dexamethasone, and Appendix A shows that an OPLS-DA model gives poor separation when patients are classified by Dexamethasone treatment (R2Y = 0.41 and Q2Y = 0.21). Therefore, Dexamethasone alone was not the reason for the separation between Wave 1 and Wave 2 patients. The separation between the two waves could be due to various reasons such as the different variants of SARS-COV-2 that were in circulation during the different collection waves, different treatment protocols and the gradual introduction of vaccinations during Wave 2. Specifically, the Wave 1 patients were recruited between May 2020 and July 2020 when the dominant variant was Wildtype, whilst Wave 2 patients were recruited between September 2020 and June 2021 when initially Wildtype dominated, with Alpha dominant from January 2020 to March 2021, following which the UK experienced the emergence of the Delta variant which was confirmed with the REACT-1 study which has been monitoring the spread and clinical manifestation of COVID-19 [24].

The data from Wave 1 and Wave 2 was combined to reveal a set of COVID-19 biomarkers that are robust enough to withstand differences in variant or treatment regime. The 6 markers identified by the machine learning model to distinguish the patients across both the waves: TG (22:1_32:5), TG (18:0_36:3), glycolithocholic acid (GLCA), glutamic acid (Glu), aspartic acid (Asp) and methionine sulfoxide (Met-SO), were shown to be significantly different between the positive (both waves) and negative patients (Figure 4). This is concordant with previous studies which have reported changes in triglycerides, bile acids and amino acids associated with COVID-19 infection [13,16,17,18,22] and are also markers of pathological effects [42,43]. Additionally, this study shows that whilst some metabolic changes vary according to the wave of infection, some changes are characteristic of COVID-19 across multiple waves. It is noteworthy that the panel of six biomarkers separating both waves achieved equivalent separation accuracy (92%) as the separation accuracy achieved for waves one (93%) and two (97%), demonstrating that sufficient features are conserved between waves to allow for accurate diagnosis.

Figure 5 shows that for the 7 patients providing longitudinal samples 2 to 7 months after the first hospital sample was taken, 2 of these markers (TG (18:0_36:3) and aspartic acid) were found to recover back to the same levels as the negatives and healthy controls. However, glutamic acid levels failed to return to levels consistent with negative patients or healthy controls. It has been widely reported that the glutamine metabolism is altered in COVID-19-positive patients [14,20,21] but this outcome suggests that the pathway is still altered months after infection. This is further supported by Bizkarguenaga et al. who showed that COVID-19 patients recruited 3 to 10 months after diagnosis had altered metabolism, particularly lipoprotein profiles, compared to individuals never infected by the virus [44]. Additionally, Wu et al. reported that lipid metabolism was altered 12 years after infection of severe acute respiratory syndrome (SARS) [45].

The study does, however, have several limitations which it is important to acknowledge. First, as a retrospective observational study conducted in a hospital setting, the sample size was constrained by the number of inpatients available for recruitment. In addition, substantially more COVID-19-positive participants were recruited in Wave 2 than in Wave 1. It was not possible in this study to confirm the variant with which the patient was infected, because the hospital RT-PCR test returned positive/negative only. Furthermore, biomarkers of COVID-19 may continue to change in the future, as new variants emerge and as population-wide immune responses improve (due to vaccination or prior infection). Future studies will be able to address these challenges via targeted and balanced recruitment across larger sample sizes thereby discovering consistent biomarker patterns.

## 5. Conclusions

This work shows that metabolite biomarkers of COVID-19 infection have a dependence on collection wave. We did however identify six metabolites that were dysregulated in COVID-19-positive patients across both waves of infection, giving a 92% classification accuracy. Univariate analysis supported this observation. Whilst two of the metabolites (TG (18:0 36:3) and Asp) were found to revert to normal levels 2–7 months after COVID-19 infection, glutamic acid remained elevated. This supports the hypothesis that some metabolic pathways are disrupted even several months after COVID-19 infection.

This work demonstrates, for the first time, that metabolic dysregulation has changed over the course of the pandemic, reflecting changes in variants, clinical presentation and treatment regimes. It also shows that some metabolic changes are consistent across waves of infection, and furthermore that despite the evolution of the virus, these can differentiate COVID-19-positive individuals from controls in a hospital setting.

## Figures and Tables

**Figure 1 metabolites-12-00713-f001:**
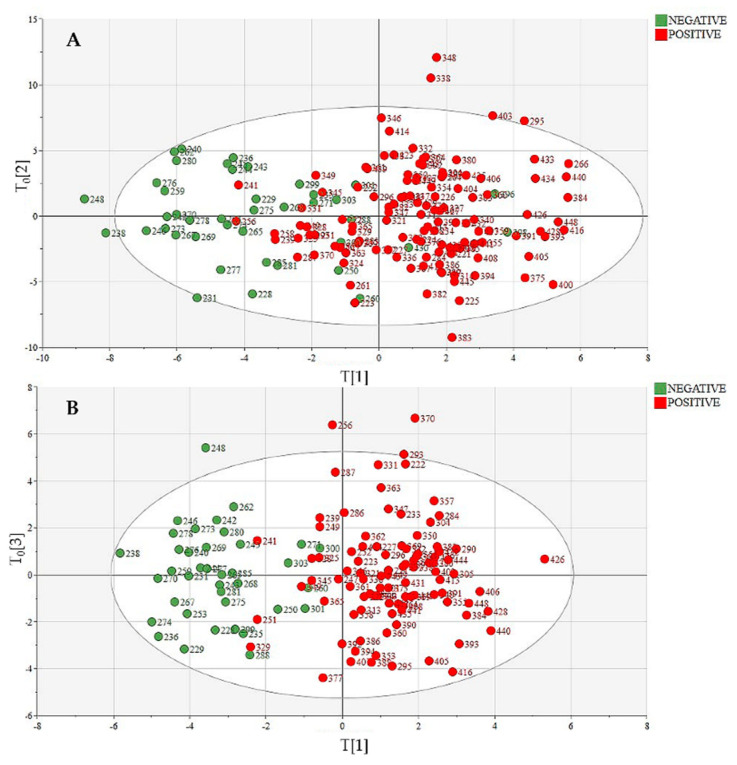
Orthogonal partial least squares discriminant analysis (OPLS-DA) of (**A**) all of the data (*n* = 164) color coded by RT-PCR test result comparing positive patients (*n* = 193) and negative patients (*n* = 41) and (**B**) only taking into account positive patients who were sampled within 14 days of the positive RT-PCR test (*n* = 97), and removing patients who were recovering from COVID-19 leaving only negative patients with no known history of COVID-19 (*n* = 37).

**Figure 2 metabolites-12-00713-f002:**
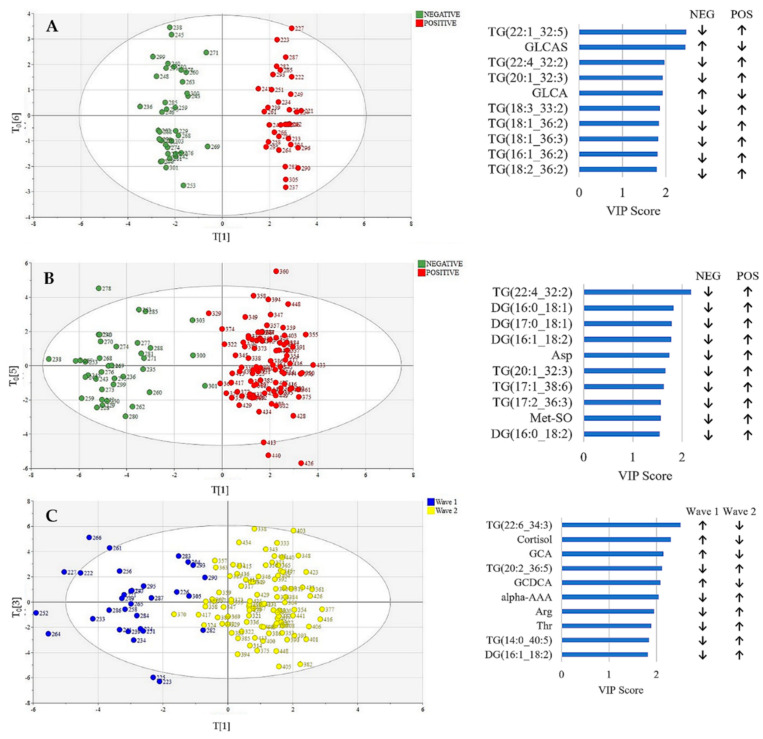
Left: Orthogonal partial least squares discriminant analysis (OPLS-DA) of (**A**) patients with positive RT-PCR tests in Wave 1 (*n* = 32) sampled from May 2020 to July 2020 compared to negative controls (*n* = 37) color coded by RT-PCR test result, (**B**) patients with positive RT-PCR tests in Wave 2 (*n* = 91) sampled from September 2020 to June 2021 compared to negative controls (*n* = 37) color coded by RT-PCR test result, and (**C**) patients with positive RT-PCR tests in Wave 1 (*n* = 32) and Wave 2 (*n* = 91) color coded by wave. Right: The top 10 VIP scores for each model, showing which metabolites are up and down regulated.

**Figure 3 metabolites-12-00713-f003:**
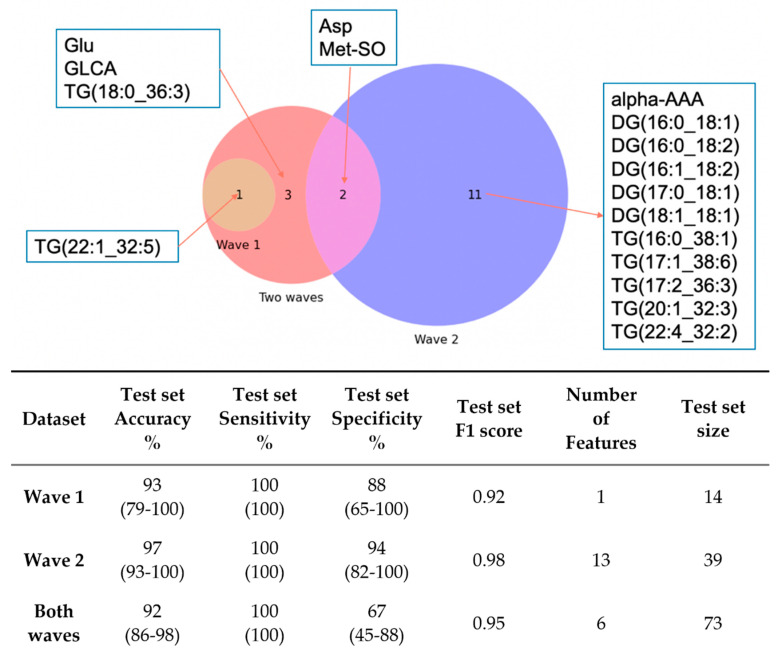
(**Top**) Venn diagram for the features identified by machine learning to be predictive for COVID-19 diagnosis in Wave 1, Wave 2 and across waves (where DG: diglycerides and TG: triglycerides); and (**Bottom**) Machine learning model prediction results in test set of Wave 1, Wave 2 and across waves, the 95% confidence intervals are enclosed in the parentheses.

**Figure 4 metabolites-12-00713-f004:**
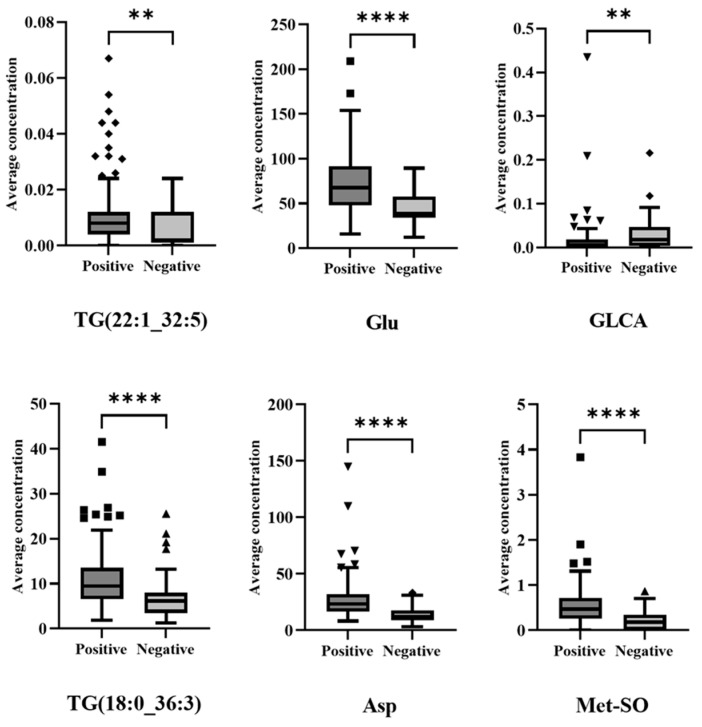
Box plots (outliers shown as symbols outside of whiskers) to compare positive and negative patients for the 6 metabolites shown to differentiate across both waves (**—*p* < 0.01, ****—*p* < 0.001).

**Figure 5 metabolites-12-00713-f005:**
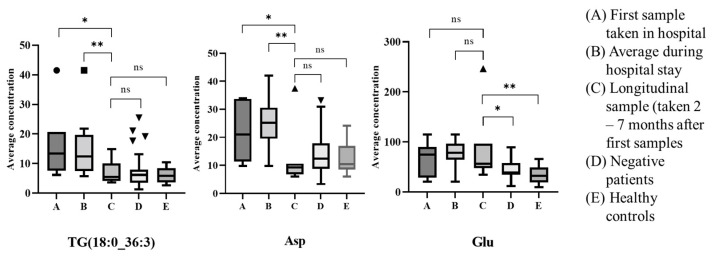
Box plots (outliers shown as symbols outside of whiskers) to show the 3 metabolites shown to differentiate across both waves for the 7 patients with longitudinal samples where (**A**) First sample taken in hospital, (**B**) average of samples over hospital stay, (**C**) longitudinal samples, (**D**) all negative patients and (**E**) Healthy controls (*—*p* < 0.05, **—*p* < 0.01, ns—not significant).

**Table 1 metabolites-12-00713-t001:** A summary of the clinical characteristics comparing positive and negative patients, and Wave 1 and Wave 2 positive patients.

	All Patients			Positive Patients		
	Negative	Positive	*p*-Value	Wave 1	Wave 2	*p*-Value
N	41	123		32	91	
Age (mean, standard deviation; years)	62.4 ± 19.9	61.6 ± 16.9	0.696	61.7 ± 19.7	61.8 ± 15.9	0.890
Male/Female (n)	16/22	81/45	0.023	17/15	64/30	0.140
Time between positive RT-PCR test and sampling (mean, standard deviation; years)	N/A	9 ± 13	-	11 ± 14	9 ± 13	0.253
Treated for Hypertension (n)	16	44	0.570	10	34	0.670
Treated for High Cholesterol (n)	7	12	0.133	5	7	0.296
Treated for Type 2 Diabetes Mellitus (n)	11	36	1.000	10	26	0.820
Treated for Ischemic Heart Disease (n)	7	19	0.618	5	14	1.000
Current Smoker (n)	1	4	1.000	2	2	0.267
Ex-Smoker (n)	12	38	0.844	4	34	0.014
Medical Acute Dependency admission (n)	7	66	0.000	12	54	0.023
Intensive Care Unit admission (n)	1	16	0.125	4	12	1.000
Did Not Survive Admission (n)	1	7	0.683	2	5	1.000
Lymphocytes (mean, standard deviation; cells/μL)	0.98 ± 0.49	0.88 ± 0.75	0.065	0.61 ± 0.34	0.96 ± 0.83	0.034
C-Reactive Protein (mean, standard deviation; mg/L)	111.30 ± 99.91	101.12 ± 102.01	0.46	164.7 ± 125.1	79.5 ± 83.1	0.000
Eosinophils (mean, standard deviation; 100/μL)	0.34 ± 0.39	0.12 ± 0.23	0.000	0.24 ± 0.36	0.07 ± 0.16	0.000
Bilateral Chest X-Ray changes (n)	5	81	0.000	18	63	0.292
Continuous Positive Airway Pressure (n)	5	44	0.014	9	35	0.397
O_2_ required (n)	12	76	0.003	17	59	0.404
Dexamethasone treatment	1	65	0.000	0	65	0.000

## Data Availability

Participant metadata data with identifiers, alongside mass spectrometry. RAW files will be made available on the Mass Spectrometry Coalition website upon publication of this study. The analytical protocols used as well as sample and participant data will be openly available for all researchers to access. The website URL is https://COVID19-msc.org/ (accessed on 28 July 2022).

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
