# Peer review of "Metabolomics Markers of COVID-19 Are Dependent on Collection Wave"

_metabolites, 2022, doi:10.3390/metabo12080713_

Round 1

Reviewer 1 Report

The article presented by Holly-May Lewis and collaborates, entitled “Targeted metabolomics shows biomarkers for COVID-19 are 2 dependent on collection wave”, is an original article that aimed to investigate metabolomic changes associated with COVID-19 infection in the first (March to June 2020) and second (September 2020 and June 2021) wave of infection in the UK.  The article is descriptive and needs to improve the results by relating the data with the clinical symptoms of the patients, differentiating between diabetics or those with hypertensive disease

Major revision:

1.       The title is complicated and convoluted, it should be improved by making it simple indicating the the main finding.

2.       It is necessary to reinforce the need/importance of the study. The sentence “The association between comorbidities such as obesity, diabetes and older age with prognostic outcomes suggests that metabolic disturbances may be significant.[7]” is not enough.

3.       The statement “Metabolic profiling therefore can identify biomarkers and unlike RT-PCR, can be used as both a diagnostic and prognostic tool which will be critical for future epidemics” has too many pretensions to only have the reference to Hutingtons disease, which is a neurogenetic disease and has nothing to do with infection with COVID 19: “Skene DJ, Middleton B, Fraser CK, Pennings JLA, Kuchel TR, Rudiger SR, et al. Metabolic profiling of presymptomatic 516 Huntington’s disease sheep reveals novel biomarkers. Sci Rep. 2017;7(February):1–16. The authors must show other and better evidence.

4.       The authors should include in the discussion some data taken from the literature the strains described with the waves studied

Reviewer 2 Report

This is an important paper, that would benefit from some minor modifications / changes.

The abstract although unstructured should contain all sections, including numerical data and final conclusion.

Although data are limited, but valuable, a sample size calculation would strengthen the power of the final conclusions. According to that. A further analysis might be needed, if a larger size would be needed.

The first paragraph of the conclusions should contain only the main and some of the secondary findings of the experimental study.

The final conclusions should have a more lay format, in order for the reader to understand clearer the real findings of the study, affected by the quality of the study presented.

Reviewer 3 Report

The article “Targeted metabolomics shows biomarkers for COVID-19 are dependent on collection wave” demonstrates differences in serum biomarkers from patients diagnosed with COVID-19 in two different time periods of the pandemic. The article presents interesting and compelling results that will be of interest to readers. I recommend that the article be accepted with minor revision.

Page 6, Table 1: Do you think there may be a bias in the study of including almost three times as many positive patients in Wave 2 compared to Wave 1?

Page 7, line 293: Instead of using “covid” use “COVID-19”.

Page 8, line 293: What were the criteria for determination of time periods for COVID-19 waves? Is there evidence to suggest the original strain for Wave 1, and then variants for Wave 2? Were samples tested to determine variants? Why was a longer timeframe chosen for Wave 2 compared to Wave 1?

Page 10: Figure 3: Label which part of the figure is A) and B). In B, it states that confidence intervals are in brackets, but I do not see this. Do you mean parentheses? Define what “DG” and “TG” metabolites stand for.

Page 13, line 426-427: Do the authors have any speculation as to what may cause altered lipid metabolism due to SARS or Covid infection, and if this has any repercussions for patients who recovered from these diseases?

Supplementary Table 1: What is MABRA ID? Suggest not including this if it could this somehow link back to patients.

Round 2

Reviewer 1 Report

The authors have improved the manuscript